# The Response of Catchment Ecosystems in Eutrophic Agricultural Reservoirs to Water Quality Management Using DOM Fluorescence

**Mei-Yan Jin [1], Jong-Jun Lee [1],\*, Hye-Ji Oh [1], Gui-Sook Nam [2], Kwang-Seuk Jeong [3], Jong-Min Oh [1],\* and Kwang-Hyeon Chang [1],\***

[1]  Department of Environmental Science and Engineering, Kyung Hee University, Yongin 17104, Gyeonggi, Korea; jinmeiyan8709@163.com (M.-Y.J.); ohg2090@naver.com (H.-J.O.)

[2]  Rural Research Institute of Korea Rural Community Corporation, Ansan 15634, Gyeonggi, Korea; leo612@ekr.or.kr

[3]  Department of Nursing Science, Dongju College, Busan 49318, Korea; kjeong@gsdongju.ac.kr

\*  Correspondence: jjoollee87@naver.com (J.-J.L.); jmoh@khu.ac.kr (J.-M.O.); chang38@khu.ac.kr (K.-H.C.); Tel.: +82-10-9300-7535 (J.-J.L.); +82-10-3316-7722 (J.-M.O.); +82-10-8620-4184 (K.-H.C.)

**Abstract:** Three-dimensional excitation emission matrix (EEM) fluorescence spectroscopy was used to investigate the characteristics of dissolved organic matter (DOM) in five typical eutrophic agricultural reservoirs. Based on catchment ecosystem, the five reservoirs were divided into three pollution sources of livestock, living, and farmland sources. The quantities and qualities of DOM in the reservoirs were analyzed. Our results showed that DOM characteristics were different in eutrophic reservoirs based on source. More protein-like components were observed in the reservoirs with the living sources, while more humic-like components were seen in the reservoir with farmland sources. Additionally, correlation analysis showed different sources for protein-like and humic-like components. Protein-like components originated mainly from phytoplankton (endogenous sources), and humic-like components were from terrestrial sources. Furthermore, the high values of specific fluorescence parameters were consistent with a dominant role of endogenous DOM in eutrophic water bodies, with FI values (fluorescence index) of approximately 1.9, and β:α values (freshness index) greater than 0.7. This result indicated that mixed features dominated endogenous sources in the reservoirs, regardless of terrestrial pollution sources. By comparing our fluorescence characteristics and historical references, we confirmed that catchment ecosystems related to human activities are important factors in determination of the characteristics of DOM in aquatic environments. However, complex and extensive eutrophication requires endogenous control of water bodies, which will play a central role in improving water environments and sustainable use of reservoirs. Therefore, this study provides an effective basis for water quality assessment of eutrophic agricultural reservoirs.

**Keywords:** dissolved organic matter (DOM); agriculture reservoir; terrestrial source; endogenous source; eutrophication

## 1. Introduction

Dissolved organic matter (DOM) is an organic continuum or mixture with different structures and molecular weights, containing abundant biogenic elements such as carbon, nitrogen, and phosphorus [1–3]. Thus, DOM plays an important role in the control of transport, speciation, and bioavailability of carbon, nitrogen, and other elements in aquatic ecosystems by providing energy sources to microorganisms [4,5], demonstrating important functions in matter cycling in aquatic ecosystems.

The compositions and characteristics of DOM in aquatic ecosystems are closely related to two main sources of terrestrial inputs and bio-endogenous release. Terrestrial DOM is transported from the rainfall, industrial and agricultural activities, and domestic sewage, while endogenous DOM sources are created by aquatic plants, plankton (phytoplankton and zooplankton), and microbial activities [6,7]. Compared to terrestrial DOM, endogenous DOM is richer in carbohydrates and more easily degraded. Therefore, endogenous DOM participates in migration and energy transfer of substances in the microbial food chain and has a significant contribution to primary productivity. In addition, interactions among biogenic elements, phytoplankton, and zooplankton play important roles in endogenous DOM [8–10]. Zooplankton gains nutrients through grazing and assimilation and release bioavailable nutrients by excretion to the water, where they are available to phytoplankton. Many studies have reported that DOM is an important source of nutrients in the algal bloom period [11,12]. Therefore, DOM could be a useful indicator for water quality management in eutrophic water based on source tracking of DOM and changes in endogenous organic pollutants [13].

The compositions and characteristics of DOM are affected by catchment ecosystem based on terrestrial sources. Studies have found that the amount of microbially derived dissolved organic matter increases when the number of farms increases and the number of wetlands decreases [14–16]. Also, a higher artificial land use rate will increase the bioavailability of DOM in water, enhancing microbial activity [14]. Therefore, catchment ecosystem is an important factor affecting the compositions and characteristics of DOM. On the other hand, DOM can be used to effectively track terrestrial sources to aid in water quality management. However, research on the fluorescence spectral characteristics of DOM in freshwater has concentrated primarily on rivers [3,5], while the influences of catchment ecosystem in eutrophic agricultural reservoirs has received relatively less attention.

The traditional purposes of agricultural reservoirs were irrigation and flood control. However, reservoirs have recently begun to play an important role not only in the sustainable water supply but also in maintaining local bio-diversity of aquatic flora and fauna [17]. In particular, reservoirs provide important temporal and available habitats for completing the life cycles of insects and amphibians. The ecological roles of reservoirs and their catchment ecosystems (surrounding landscapes) are also important. Therefore, water quality management of agricultural reservoirs is important because each reservoir has a variety of sources that decrease water quality or provide food sources for the aquatic food web. The traditional physicochemical parameters represent only quantitative changes of nutrients in freshwater, which are difficult to use to explore the endogenous responses of water quality to terrestrial pollution inputs. DOM fluorescence was reported to be a useful technique for estimating and managing water quality because it can represent the changes of DOM composition and indicate pollution sources [18]. Further, recent research provides evidence of significant correlations between DOM and traditional physicochemical parameters [16,19]. Hence, DOM could be an important biogeochemical indicator for analyzing eutrophic agricultural reservoirs. Three-dimensional excitation emission matrix (EEM) fluorescence is a simple, rapid, sensitive, informative, and low-cost method that can provide a reliable tool to measure the structural features of DOM to understand its transformations in aquatic ecosystems [4,20]. In this study, we used EEM to characterize eutrophic agriculture reservoirs. DOM fluorescence values of five reservoirs were used to characterize the influence of three typical catchment ecosystems. The main objective of this study was to characterize the endogenous sources and explore the effects of terrestrial pollution sources on water quality based on composition and characteristics of DOM for effectively estimating and managing the water quality. This information can serve as a reference for understanding and developing better management practices of agricultural reservoirs.

## 2. Materials and Methods

### 2.1. Sampling Sites

We selected five eutrophic agriculture reservoirs in South Korea based on previous reports [21,22]. These reservoirs were selected because they had different dominant pollution. Although all are affected

by farmland and human activities (living and livestock sources) because they are agricultural reservoirs, the dominant pollution sources are different among the reservoirs. The percentages of dominant pollution sources to total pollution load to each reservoir were calculated according to nitrogen and phosphorus inputs [21,22]. Bongrim (BR) and Yidam (YD) reservoirs were polluted by livestock sources at percentages of 59% and 50%, respectively. Hongjung (HJ) and Bongam (BA) reservoirs were affected by domestic sewage, and the HJ reservoir was polluted at a percentage of 86%. Nanjung reservoir (NJ) was affected by farmland by up to 94%. These reservoirs are typical eutrophic reservoirs, in which the dominant phytoplankton species were identified as *Microcystis aeruginosa* by microscopy in the laboratory. During the sampling period, we observed that inflows and outflows from the reservoir were very limited; therefore, it is assumed that their residence times were long. Basic specifications such as location, year of installation, and other parameters were supplied by the Rural Research Institute of Korea Rural Community Corporation (Table 1). The effective storage capacities of the reservoirs were different.

**Table 1.** Location of sampling sites (Yidam, Bongrim, Hongjung, Bongam, and Nanjung) and basic reservoir structure properties.

| Reservoir | Yidam (YD) | Bongrim (BR) | Hongjung (HJ) | Bongam (BA) | Nanjung (NJ) |
|---|---|---|---|---|---|
| Longitude and latitude | 36°51′21.6″ N 127°52′46.9″ E | 36°43′49.7″ N 126°39′19.4″ E | 37°24′01.7″ N 126°44′26.9″ E | 37°55′03.7″ N 126°59′49.6″ E | 37°47′03.0″ N 126°13′40.8″ E |
| Pollution source | livestock | livestock | living | living | farmland |
| Installation year | 1931 | 1944 | 1957 | 1979 | 2006 |
| Basin area (ha) | 535 | 840 | 710 | 340 | 1884 |
| Benefit area * (ha) | 133 | 230 | 127 | 112 | 889 |
| Effective storage capacity ($10^3$ $m^3$) | 644 | 1065 | 483 | 967 | 6214 |
| Mean depth (m) | 2.37 | 9.33 | 3.23 | 10.72 | 2.70 |

* The benefit area is the area that benefited from the reservoir (such as farmland).

## 2.2. Sampling Strategy and Physico-Chemical Analyses

Water samples from the surface of the pelagic zone of each reservoir (the center of the reservoir) were collected monthly from May to October 2018 during the daytime (10:00–15:00). The sampling and measurement of basic water quality at each reservoir took approximately two hours. The water samples were filtered through 0.45 μm glass microfiber filters (Whatman, GF/F, 47 mm in diameter) before analysis of dissolved carbon (DOC) and DOM fluorescence in situ. The water samples were stored in a laboratory at 4 °C for less than 48 h. The temperature, pH, and electrical conductivity (EC) of each water sample were measured onsite with a Horiba U-50 Series multiparameter water quality meter. In the laboratory, DOC was measured with a Shimadzu TOC–VCSH total organic carbon analyzer. Data on mean depth, basin area, effective storage capacity, installation year, and water pollution source of the reservoirs for structural analysis, total nitrogen (TN), total phosphorus (TP), and chlorophyll a (Chl a) were supplied by the Rural Research Institute of Korea Rural Community Corporation. TN was measured with the alkaline persulfate digestion method, TP was quantified with the ammonium molybdate method [23], and Chl a was measured by the acetone extraction method [24].

## 2.3. EEM Fluorescence Spectroscopy

Before the EEM measurements, the samples were diluted with Milli-Q filtered water until the ultraviolet absorbance at 254 nm was below 0.05 $cm^{-1}$ [25] to avoid inner filter correction. The pH was fixed at 3.0 for fluorescence measurements to minimize potential interference from metals present. Three-dimensional EEM of DOM samples from five reservoirs were analyzed using a Hitachi Fluorescence Spectrophotometer (F-7000) equipped with the Fluorescence Solutions 2.1 software [26,27] for data processing. Fluorescence scans were conducted at excitation wavelengths from 200 to 450 nm using 5 nm increments and emission wavelengths from 280 to 550 nm at 5 nm increments, a bandwidth of 5 nm, and 0.5 s of integration time. Milli-Q filtered water was used as a blank. The inner filtering

effect was corrected by absorbance spectroscopy [28]. Fluorescence intensities were normalized in the area under the Milli-Q water Raman peak and converted to quinine sulfate units [9].

Fluorescence regional integration (FRI) was applied to quantify the EEM results, which were divided into five fluorescence regions of DOM by wavelengths of excitation emission [1,29,30] (Table 2). To reflect the organic relative amount of special structures in the specific region, the cumulative excitation emission area volumes were normalized to relative regional areas. The normalized excitation emission area volumes ($\Phi_{i,n}$, referring to the value of region i) and the percent of fluorescence response ($P_{i,n}$) were calculated by OriginPro 2016 software. The available indexes calculated in this study were fluorescence index (FI), humification index (HIX), biological index (BIX), and freshness index ($\beta{:}\alpha$). The FI value is the ratio of fluorescence intensities of emission wavelengths of 470 and 520 nm (excitation was kept at 370 nm) [31]. The HIX value is the ratio of the integral area of emission wavelength region of 435–480 nm and 300–345 nm at constant excitation of 254 nm [32,33]. The BIX value is the fluorescence intensity ratio at emission wavelengths of 380 and 430 nm and excitation of 310 nm [34]. The $\beta$ and $\alpha$ values are calculated as excitation at 310 nm based on emission intensity at 380 nm ($\beta$ region) divided by the emission intensity maximum between 420 and 435 nm ($\alpha$ region) [15,35].

**Table 2.** Major fluorescent components of dissolved organic matter (DOM) in five reservoirs.

| Region | Ex (nm) | Em (nm) | Component Type |
|:------:|:-------:|:-------:|----------------|
| B | 225–290 | 300–320 | Tyrosine-like |
| T | 225–290 | 320–370 | Tryptophan-like |
| A | 225–290 | 370–460 | Fulvic acid-like |
| D | 290–360 | 300–370 | Soluble microbial by product-like (SMB) |
| C | 290–440 | 370–530 | Humic acid-like |

## 3. Results

### 3.1. Physico-Chemical Characteristics of Reservoir Water

The variations in water quality were associated with a noticeably eutrophic condition in the reservoirs (Table 3). The pH of most reservoirs ranged from 7.19 to 10.77, revealing temporal changes, increased values from June to August, and decreased values from September to October. The EC values in the NJ reservoir, for which farmland and soil erosion were pollution sources, were higher than those in other reservoirs. TN concentrations showed strong temporal variation and had peaks in May and then sharply decreased from June to August in all reservoirs. Relatively low TN concentrations were observed in the BA and NJ reservoirs, with reading approximately half values to those of other reservoirs. However, TP followed an inverse temporal variation to TN, with an increasing pattern occurring from July to September and minimal values occurring in October. TP concentrations in most reservoirs (except YD) ranged from 0.03 to 0.10 mg/L, with an average of 0.05 mg/L. The average Chl a concentrations in all reservoirs ranged from 36.15 to 63.02 μg/L, which indicated the waters were in an algal bloom period. A continuing outbreak of algal bloom occurred from May to October, with peak concentrations of Chl a in September and October (mean value: 142.38 μg/L). The lowest concentration of Chl a was observed in the BA reservoir (98.90 μg/L) in September, while the highest concentration was observed in the YD reservoir (188.30 μg/L) in October. DOC concentrations in all reservoirs ranged from 3.23 ± 1.09 to 6.19 ± 1.12 mg/L (mean value ± standard deviation), and further to 9.29 ± 1.39 mg/L. DOC also showed strong temporal variations, with increasing patterns from June to August and relatively high concentrations in September and October (similar to the Chl a patterns).

**Table 3.** Basic characteristics of water samples in five reservoirs (Yidam, Bongrim, Hongjung, Bongam, and Nanjung).

| Reservoir | YD | BR | HJ | BA | NJ |
|---|---|---|---|---|---|
| | Range | Range | Range | Range | Range |
| | Mean (± SD) | Mean (± SD) | Mean (± SD) | Mean (± SD) | Mean (± SD) |
| Temp (°C) | 20.50–34.85 | 21.10–34.81 | 21.79–30.39 | 19.99–31.03 | 19.47–33.47 |
| | 26.80 ± 5.63 | 28.03 ± 4.94 | 26.87 ± 3.61 | 26.35 ± 4.02 | 26.26 ± 5.89 |
| pH | 7.19–10.77 | 8.32–10.43 | 8.67–10.12 | 7.79–10.42 | 7.31–10.05 |
| | 9.46 ± 1.57 | 9.74 ± 0.79 | 9.33 ± 0.56 | 9.43 ± 0.90 | 8.82 ± 1.12 |
| EC (mS/cm) | 0.11–0.25 | 0.14–0.16 | 0.22–0.28 | 0.18–0.23 | 0.35–0.54 |
| | 0.18 ± 0.06 | 0.15 ± 0.01 | 0.25 ± 0.02 | 0.21 ± 0.02 | 0.45 ± 0.08 |
| TN (mg/L) | 0.46–3.40 | 1.62–2.90 | 1.59–3.15 | 0.74–1.37 | 0.85–2.29 |
| | 2.18 ± 1.17 | 2.12 ± 0.53 | 2.53 ± 0.54 | 1.07 ± 0.27 | 1.42 ± 0.53 |
| TP (mg/L) | 0.07–0.39 | 0.04–0.06 | 0.04–0.09 | 0.03–0.07 | 0.03–0.10 |
| | 0.16 ± 0.12 | 0.05 ± 0.01 | 0.06 ± 0.02 | 0.05 ± 0.02 | 0.05 ± 0.03 |
| Chl a (ug/L) | 16.30–188.30 | 20.50–132.70 | 16.10–116.50 | 11.40–98.90 | 6.30–175.50 |
| | 63.02 ± 67.27 | 52.98 ± 41.15 | 37.63 ± 38.98 | 36.15 ± 33.82 | 58.68 ± 65.60 |
| DOC (mg/L) | 3.29–9.14 | 2.53–7.38 | 2.29–9.75 | 2.73–8.63 | 5.32–11.54 |
| | 5.94 ± 2.55 | 4.90 ± 2.19 | 6.22 ± 2.84 | 5.63 ± 2.17 | 8.24 ± 2.05 |

Note: EC, electrical conductivity; TN, total nitrogen; TP, total phosphorus; Chl a, chlorophyll a; DOC, dissolved organic carbon.

## 3.2. FRI Components of DOM

The excitation emission area volumes of the five components showed strong temporal variations in all reservoirs, and the differences among reservoirs were significant (Figure 1). The integral values of DOM components increased from June to August following a pattern similar to that of DOC. The dominant components of DOM in the reservoirs were fulvic acid-like ($\Phi_{A,n}$: $0.93 \pm 0.18 \times 10^3$) and tryptophane-like ($\Phi_{T,n}$: $0.76 \pm 0.22 \times 10^3$). Tyrosine-like ($\Phi_{B,n}$: $0.28 \pm 0.06 \times 10^3$) and the soluble microbial byproduct-like ($\Phi_{D,n}$: $0.22 \pm 0.09 \times 10^3$) components were the next most common, and the humic acid-like ($\Phi_{C,n}$: $0.19 \pm 0.03 \times 10^3$) component was the least. According to the different types of reservoirs (grouping by different pollution sources), and with the exception of the tryptophan integral value ($\Phi_{B,n}$), the orders of the protein components ($\Phi_{T,n}$ and $\Phi_{D,n}$) were living sources (HJ and BA) > farmland sources (NJ) > livestock sources (BR and YD). The humus components ($\Phi_{A,n}$ and $\Phi_{C,n}$) are shown as farmland sources (NJ) > living sources (HJ and BA) > livestock sources (BR and YD). The amounts of each component, measured as percent of $\Phi_{i,n}$ for the five components ($P_{i,n}$), varied substantially among all DOM fractions (Figure 2). The proportions of the five components were similar in all reservoirs. $P_{B,n}$ ranged from 10.00% to 14.07%, $P_{T,n}$ from 27.88% to 33.86%, $P_{A,n}$ from 37.08% to 44.28%, $P_{D,n}$ from 7.43% to 9.93%, and $P_{C,n}$ from 7.21% to 9.81%.

To further investigate the fate of different components of DOM in the five reservoirs, a correlation analysis was conducted (Table 4). Among the five reservoir samples (except HJ), the humus components (A and C) had no significant correlation with the protein-like components (B, D, and T). However, the A and C components were significantly correlated ($p < 0.01$) (except YD), and the correlation between protein-like components was also significant ($p < 0.01$). These results showed that the sources of the proteins and humus components were different in four of the five reservoirs (except HJ), while the humus components represented by components A and C had the same sources, and the protein-like components represented by B, T, and D also had the same sources (except YD).

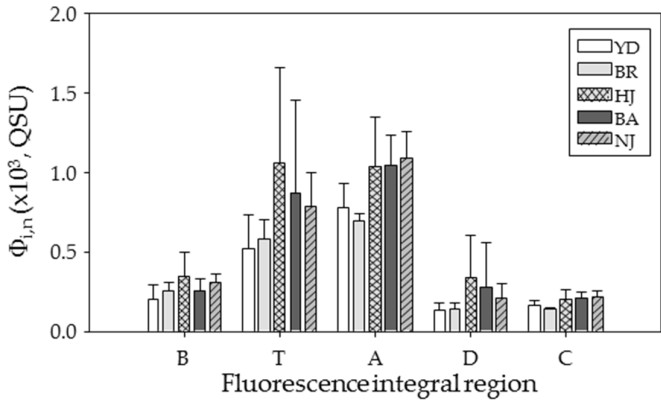

**Figure 1.** Distributions of the abundance of DOM components in the five reservoirs.

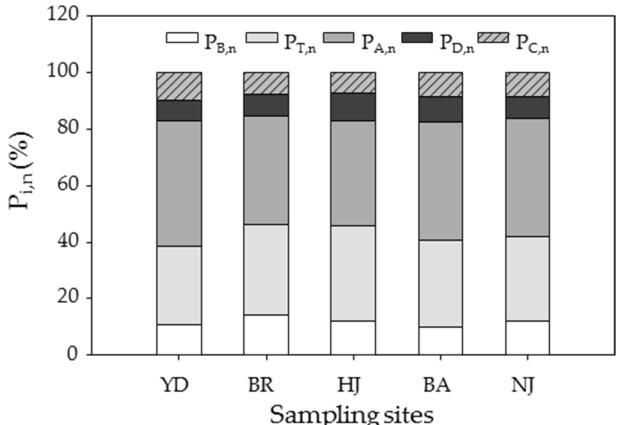

**Figure 2.** Distributions of the abundance of DOM components in the five reservoirs.

**Table 4.** Correlation among different fluorescence components of DOM from different reservoirs (Yidam, Bongrim, Hongjung, Bongam, and Nanjung).

| | YD | | | | | | BR | | | | |
|---|---|---|---|---|---|---|---|---|---|---|---|
| Fluorescence Components | B | T | A | D | C | Fluorescence Components | B | T | A | D | C |
| B | 1.00 | | | | | B | 1.00 | | | | |
| T | 0.99 ** | 1.00 | | | | T | 0.94 ** | 1.00 | | | |
| A | 0.69 | 0.70 | 1.00 | | | A | −0.18 | −0.20 | 1.00 | | |
| D | 0.90 * | 0.94 ** | 0.57 | 1.00 | | D | 0.81 | 0.95 ** | −0.17 | 1.00 | |
| C | −0.06 | −0.04 | 0.68 | −0.13 | 1.00 | C | −0.66 | −0.68 | 0.83 * | −0.60 | 1.00 |

| | HJ | | | | | | BA | | | | |
|---|---|---|---|---|---|---|---|---|---|---|---|
| Fluorescence Components | B | T | A | D | C | Fluorescence Components | B | T | A | D | C |
| B | 1.00 | | | | | B | 1.00 | | | | |
| T | 0.80 | 1.00 | | | | T | 0.96 ** | 1.00 | | | |
| A | 0.96 ** | 0.68 | 1.00 | | | A | 0.61 | 0.61 | 1.00 | | |
| D | 0.67 | 0.98 ** | 0.54 | 1.00 | | D | 0.95 ** | 0.99 ** | 0.57 | 1.00 | |
| C | 0.93 ** | 0.62 | 0.96** | 0.47 | 1.00 | C | 0.52 | 0.50 | 0.98 ** | 0.44 | 1.00 |

| | NJ | | | | |
|---|---|---|---|---|---|
| Fluorescence Components | B | T | A | D | C |
| B | 1.00 | | | | |
| T | 0.85 * | 1.00 | | | |
| A | 0.35 | 0.48 | 1.00 | | |
| D | 0.80 | 0.99 ** | 0.37 | 1.00 | |
| C | 0.22 | 0.35 | 0.99 ** | 0.25 | 1.00 |

* Correlation is significant at the 0.05 level (2-tailed). ** Correlation is significant at the 0.01 level (2-tailed).

*3.3. Fluorescence Characteristic Parameter*

The FI is widely used to distinguish DOM. When FI > 1.9, DOM is derived mainly from the biological activities of microorganisms and algae, based largely on the characteristics of endogenous sources, while FI < 1.4 suggests a dominant terrestrial, higher-plant DOM source [31,34]. Overall, the two sources can be distinguished by FI values (Figure 3a). When the FI values of the reservoirs are between 1.4 and 2.0, the DOM contained a contribution from both terrestrial and endogenous DOM. Clearly, a mixture of sources was common in the reservoirs. Still, the FI mean values of the reservoirs ranged from 1.79 to 1.98, which is close to the critical point (1.9), indicating that the endogenous contribution was more dominant. A HIX value > 16 signals DOM with strong humification characteristics and indicates a primarily terrestrial contribution; a range between 6 and 10 indicates strong humification characteristics and a weaker endogenous contribution; between 4 and 6 indicates weaker humification and endogenous sources; and a value less than 4 indicates endogenous sources are dominant [33,34]. As shown in Figure 3, the average HIX values of the five reservoirs ranged from 2.07 to 3.91, indicating mainly endogenous sources. Studies have shown that BIX values reflect the relative endogenous contribution of DOM and are also used to evaluate bioavailability. A BIX value > 1 indicates an endogenous source related to organisms or bacteria; values 0.8 and 1.0 are associated with a large amount of endogenous DOM in the sample; and when the value is 0.6–0.8, the influence of endogenous DOM is less than that of terrestrial DOM [36]. The average BIX values of the five reservoirs were between 0.99 and 1.41, which confirms high relative endogenous contributions to DOM in the eutrophic reservoirs (Figure 3b). Overall, DOM in the five reservoirs was dominated by endogenous DOM, and the contribution of new endogenous DOM was large.

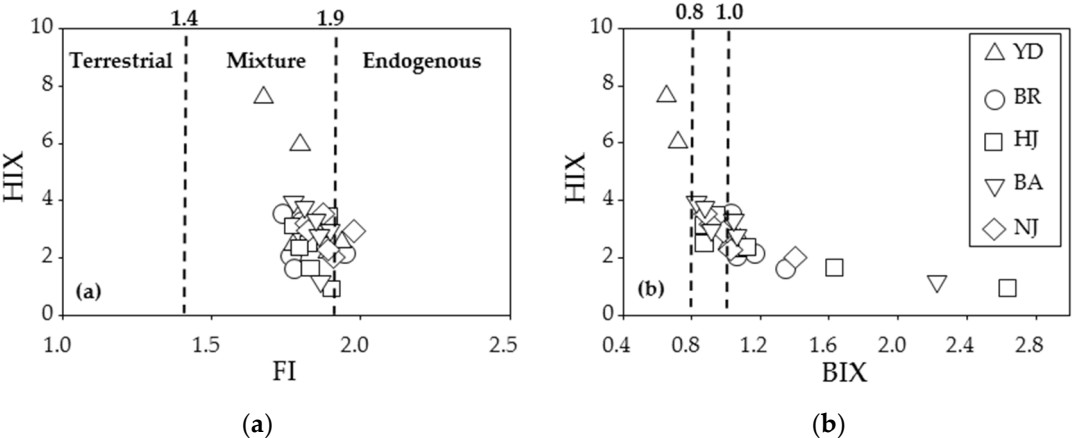

**Figure 3.** Distributions of FI versus HIX (a), and BIX versus HIX (b) for five reservoirs. (FI indicate Fluorescence index, HIX indicate Humidification index, and BIX indicate Biological index).

## 4. Discussion

The typical sources of nitrogen and phosphorus in eutrophic lakes during summers include terrestrial input (domestic sewage and industrial and agricultural wastewater), atmospheric deposition, algal degradation, and internal sediment release [37]. The results of TN, TP, and Chl a varied widely (Table 3), and the patterns were similar to those of typical eutrophic lakes [37–40]. The temporal variations of TN in the five reservoirs from May to October showed a similar trend to other research [37,38] with a decreasing pattern during the algal bloom period. This finding assumes that algal succession and growth are related to availability of nitrogen and that nitrogen was the probable limiting nutrient for *Microcystis* [12,19,37]. For clarification, concomitant nutrients and algal succession should be looked for in future complementary assessments. TP values showed an inverse pattern to TN, with a high mean value close to 0.05 mg/L and typical eutrophication [37,38]. In most instances, nutrient enrichment enhanced algal growth (as measured by Chl a) and bloom potential [12,41,42].

Chl a concentration remained relatively high in the reservoirs and was associated with TN and TP variations [38]. The two important factors for variations of TN, TP, and Chl a were sediment release with increasing temperature and uninterrupted algal degradation in the algal bloom period [37]. Additionally, enhanced microbial activity and algal growth may have contributed to increasing DOC in the warm season [18], as DOC values in all reservoirs increased in summer and were consistent with Chl a measurements. Because the eutrophic agriculture reservoirs represented three different catchment ecosystems in this study, TN concentrations of farmland sources were lower than those of living and livestock sources, Chl a concentration of living sources was lower than those of livestock and farmland sources, and DOC concentrations of the farmland source were higher than those of livestock and living sources. These findings show a clear relationship between water quality and pollution source. The high concentrations and temporal and spatial variations in the reservoirs can be explained as follows: (1) the reservoirs exhibit low flow and long residence time (unlike rivers); (2) the reservoirs are affected by many pollution sources; and (3) the reservoirs have been in a state of eutrophication for an extended period.

The variations of DOM components are associated with water quality [16,19], and the integral values of the five DOM components in all reservoirs showed significant temporal variations (Figure 1). The increasing trends were due to increasing air temperature (mean value: 22 °C, maximum temperature: 28 °C) [43], with algal bloom observed with increasing temperature and abruptly decreasing in September with loss of eutrophication. Previous studies of surface waters have shown that the amount of DOM measured as the concentration of DOC [44] and the temporal patterns of DOM components in this study were similar to the DOC results. These findings are supported by the significant relationship between DOM and water quality [16]. Furthermore, analysis of eutrophic agriculture reservoirs with three different pollution sources showed that the protein components ($\Phi_{T,n}$ and $\Phi_{D,n}$) were higher in those with living sources (HJ and BA). This finding indicates that discharge of domestic wastewater from surrounding residents has a greater influence on protein content than direct input of farmland and livestock sources [40]. The humus components ($\Phi_{A,n}$ and $\Phi_{C,n}$) were high with farmland sources (NJ), indicating that runoff input from the surrounding farmland has a greater impact on humus content than other inputs [14–16]. For the dominant components of DOM in eutrophic reservoirs, tryptophan-like substances in the lake originated mainly from residues of phytoplankton [45] and were exuded directly by live algae [12,46]. These findings suggested that the observed fluorescence components from the reservoirs were from terrestrial and endogenous sources.

For five eutrophic agriculture reservoirs with three pollution sources, the humus components (A and C) were co-sourced in most, and the order of correlation coefficients was NJ (r = 0.99) > HJ and BA (r ≈ 0.97) > BR (r = 0.83), indicating that the degree of humus development and structure of DOM were affected by pollution source. NJ reservoir is surrounded by farmlands, where the degree of soil humus development is lower than that of a forest ecosystem [47], and the structure and composition of the humus input are simpler than those of living and livestock sources. In contrast, BR reservoir is surrounded by livestock sources (dominant pollution sources) [21,22] and both farmland and woodland ecosystems, resulting in more complicated DOM structures than in those with farmland sources. The protein-like components (B and T) were also co-sourced, and the order based on correlation coefficient was YD and BR (r ≈ 0.97) > BA (r = 0.96) > NJ (r = 0.85), indicating that direct input from farmlands had a greater impact on the structure of protein-like components compared with discharge of domestic wastewater and pollution caused by animal husbandry. Therefore, the influences of the three catchment ecosystems on terrestrial DOM components were reflected to the same extent in the correlation analysis of humus and protein-like components.

The indexes of FI, HIX, and BIX further and significantly explained the characteristics and sources of DOM [48–52]. The FI values of the reservoirs classified according to pollution sources were ranked as YD and BR (livestock, FI ≈ 1.81) < BA and HJ (living, FI ≈ 1.84) < NJ (farmland, FI = 1.98) (Figure 3a). The five eutrophic agriculture reservoirs showed dominant endogenous pollution sources, with livestock and living sources contributing slightly to terrestrial pollution. These results were supported by two

previous findings. One is the algal bloom period when the produced amount of endogenous sources with algal degradation [6,7]. The other is terrestrial sources with different catchment ecosystems around eutrophic reservoirs allowing nitrogen and phosphorus to enter the aquatic ecosystems with runoff, increasing primary productivity and enhancing the endogenous source [49–52]. These findings show that endogenous DOM is an important source of nutrients during the algal bloom period; however, for accurate assessment and management of the agriculture eutrophic reservoirs, further analyses of endogenous and terrestrial pollution sources during nonbloom periods are necessary for better understanding the temporal dynamics of DOM and nutrients and their consequent impacts on reservoir water quality. The relative order of concentrations of Chl a in the five eutrophic reservoirs was living sources (≈36.89 μg/L) < livestock sources (≈53.00 μg/L) < farmland sources (=58.68 μg/L) (Table 3), providing further evidence that the DOM of the reservoirs was dominated by the endogenous DOM of algae metabolism and was an important source of protein and fulvic acid components. Studies have shown that FI values are negatively correlated with DOM aromaticity; the higher the FI value, the weaker aromaticity and the lower degree of humification [34], which is consistent with the results of the present study. This finding supports endogenous pollution sources of the reservoirs. The HIX mean value of the five reservoirs was 2.87 ± 0.64 (less than 4), and the BIX mean value of the five reservoirs was 1.17 ± 0.16 (more than 1), which further indicate that higher relative endogenous contributions of DOM showed in the eutrophic reservoirs (Figure 3) [45–49]. The metabolic activities of microorganisms and algae were relatively strong, and the results are consistent with the $\beta$:$\alpha$ values (mean value range of the $\beta$:$\alpha$ ratio of the five reservoirs was 0.96–1.34) (Table 5), indicating primarily endogenous DOM sources of the reservoirs [32].

**Table 5.** Comparison of different types of lakes and reservoirs.

| Reservoir and Lake | FI | BIX | HIX | $\beta$:$\alpha$ ratio | Pollution Source | Reference |
|---|---|---|---|---|---|---|
| Yidam (YD) | 1.68 ~ 1.94 | 0.74 ~ 1.16 | 2.16 ~ 7.54 | 0.73 ~ 1.09 | livestock | This study |
| Bongrim (BR) | 1.72 ~ 1.95 | 0.98 ~ 1.35 | 1.61 ~ 3.51 | 0.94 ~ 1.23 | livestock | "* |
| Hongjung (HJ) | 1.78 ~ 1.90 | 0.95 ~ 2.65 | 0.87 ~ 3.34 | 0.91 ~ 2.48 | living | " |
| Bongam (BA) | 1.77 ~ 1.89 | 0.91 ~ 2.26 | 1.15 ~ 3.92 | 0.87 ~ 2.15 | living | " |
| Nanjung (NJ) | 1.82 ~ 1.98 | 0.96 ~ 1.46 | 1.91 ~ 3.43 | 0.94 ~ 1.39 | farmland | " |
| Albufera des Grau | 1.39 | 0.63 | 8.93 | - | farmland and forest | [49] |
| Lumpen | 1.28 | - | 0.97 | 0.41 | forest | [50] |
| Paldang | 1.9 | - | 3.50 | - | farmland | [51] |
| Valloxen | 1.40 | - | 0.90 | 0.62 | farmland | " |
| Morii | 1.25 | 1.10 | 2.21 | - | living | [52] |
| Moghioros | 1.29 | 1.11 | 0.64 | - | living | " |
| Circului | 1.25 | 0.96 | 1.87 | - | living | " |

* " indicates the same reference as above.

Compared with previous studies, it is evident that most endogenous DOM inputs were replenished mainly by surface runoff, but some lakes were also affected by industrial and agricultural sewage, domestic wastewater, and rivers entering the lake (Table 5) [48–52]. In Albufera des Grau Lake, Valloxen Lake, and Lumpen Lake [49,51], which are surrounded by farmland and forests, FI values ranged from 1.28 to 1.40, suggesting that the DOM contained contributions from terrestrial sources. However, opposite results were seen in NJ reservoir (farmland source), with high FI values approaching 1.9, $\beta$:$\alpha$ values above 1.0, and low HIX values less than 8.0, which indicated greater endogenous sources, rapid biological metabolism, and high new organic matter content of algae. The results are consistent with data from Paldang Lake [51]. The DOM sources for HJ and BA reservoirs were endogenous and opposite those of Morii Lake, Circului Lake, and Moghioros Lake [52], which were influenced by human factors such as urban land use and sewage. The FI values were approximately 1.9, which were consistent with that of eutrophic water. Results of the analysis of YD and BR reservoirs (affected by livestock) are similar to those of HJ and BA reservoirs (affected by living). Hence, this and other studies showed that catchment ecosystem can significantly affect the reservoirs and can be effectively identified using DOM fluorescence.

## 5. Conclusions

Three-dimensional excitation emission matrix fluorescence spectroscopy combined with multivariable analysis provided excellent results for the structural composition of DOM and to track pollution sources in five eutrophic reservoirs comprising three catchment ecosystems. DOM components showed strong temporal variations and similar patterns to those of Chl a and DOC, indicating a significant relationship with water quality in the eutrophication period. DOM was dominated by tryptophan-like and fulvic acid-like components and showed high protein-like content and low humification level, indicating that endogenous DOM was the major contributor to water quality deterioration in the eutrophication period. The catchment ecosystems representing three terrestrial sources showed minor changes among the five reservoirs in the eutrophication period. Reservoirs with living sources (HJ and BA) showed high values of protein-like components, and that with farmland sources (NJ) showed high humic components. Therefore, we concluded that DOM composition can be used as an effective indicator to track pollution sources to develop assessment and management practices, improve water quality, and ensure long-term sustainability of agriculture reservoirs.

**Author Contributions:** Conceptualization, M.-Y.J. and K.-H.C.; methodology, M.-Y.J. and K.-H.C.; formal analysis, M.-Y.J. and K.-S.J.; investigation, H.-J.O. and K.-H.C.; writing—original draft preparation, M.-Y.J. and K.-H.C.; writing—review and editing, K.-H.C., J.-J.L., and J.-M.O.; visualization, M.-Y.J., J.-J.L., and K.-H.C.; supervision, K.-H.C. and J.-M.O.; project administration, G.-S.N.; funding acquisition, J.-M.O.

**Funding:** This research was funded by Korea Environment Industry & Technology Institute (KEITI) by the Korea Ministry of Environment (MOE) as Advanced Water Management Research Program (98640).

**Conflicts of Interest:** The authors declare no conflict of interest.

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
