# Peer review of "The Response of Catchment Ecosystems in Eutrophic Agricultural Reservoirs to Water Quality Management Using DOM Fluorescence"

_sustainability, doi:10.3390/su11247207_

Round 1

Reviewer 1 Report

This manuscript studied the characteristics of DOM in five typical eutrophic agricultural reservoirs using EMM fluorescence spectroscopy.

The title is confusing. Why did the authors use catchment ecosystems not reservoirs? Why used water-quality management not catchment land uses?   

The purpose statement of this MS is not clear.

What are the relationships between fluorescence parameters and nutrients? What are possible endogenous controls of these DOM in these reservoirs?

Specific comments:

Line 16-17: The characteristics of DOM ….. It is an awkward sentence.

Line 36: It is an important …?

Line 57: .. in in?

Line 56-57: This sentence is not clear.  …. affect the composition and characteristics of the main DOM sources ?

Line 62: providing important ……habitats for life cycle completion of insects and amphibians.

Line 63: from a cultural …. ?

Line 185: When …. between 1.4 and 2.0, …?

Line 197: >1indicate, add space.

Line 198: ab endogenous ..?

Line 292-293: Please specify how endogenous sources or terrestrial control are related to reservoir management.

Reviewer 2 Report

Keywords: I suggest adding “reservoir” and “agriculture”

Global comments on the paper:

The paper focuses on the characterization of Dissolved Organic Matter in lake reservoirs, and links that with the main characteristics of their watersheds. The paper is valuable and interesting. But it lacks some dedicated data to demonstrate that the results from DOM fluorescence are consistent with the reservoirs and their catchments specificities. You’ll find hereafter my comments:

Global: The English should be revised. Lines 20, 21, 23: You should avoid using abbreviation in the abstract that are not defined in the abstract. Someone who read the abstract should understand it without being obliged to read the full paper. Line 31: I don't think that “management” is the good word to use. I suggest using “assessment” Line 36: I don’t think that DOM is mainly characterized by a high level of oxygen. It is the reverse, as it needs a lot of dissolved oxygen to being mineralized. Lines 50-52: Why "studies" whereas you indicate only one single reference? Line 57: double “in”. Line 65: suppress “the” Line 85 table 1: What do you mean by “benefit area”? Line 85 Table 1: You should add much more details such as the mean residence time, the watershed surface partition per activities, the flux entering in the lakes… Lines 87-88: What about other seasons? This could be an essential issue linked with the residence time of the reservoirs. Lines 87-88: What is the location of the sampling sites (how far is it from the banks)? Even more important is the sampling frequency, and the hours of sampling, that impact both primary production and pH, then nutrient cycles. Line 197: Add a blank after “1” Line 207: There are many publications on phosphorus and nitrogen in lakes, not only in lake Tahoe. You could add more references. Line 209: You cannot conclude on that, because you have not surveyed late autumn, winter and early spring seasons. Moreover, you should also look at algal succession to be able to conclude. Finally, you don't indicate the measurement frequency, that could be a key impact factor. Lines 209-211: I think you are wrong. It directly impacts algal species, but not necessarily algal biomass. It is different than your assertion. Line 211: Could you clarify if it is due to rainy season or to internal sources? Lines 218-219: “Weak water fluidity” is not the right wording. Moreover, what is the mean residence time in the 5 lakes? it is a key data that you have not provided. Line 227: You should assess the link with meteorological parameters. Line 229: Your references are specifically linked with microcystis aeruginosa. Do you have any idea of the main phytoplankton species of your 5 reservoirs? Line 236: What do you mean by "living" not included in livestock? Lines 291-294: I think you are wrong in your conclusion. Practically, if you have a high endogenous source, it should be due to a high terrestrial source of nutrients (and/or to a high internal loading), leading to a significant biomass in a reservoir. So, limitation of terrestrial inputs is always a key issue. Global comment: The main issue is that you have not clear data on the 5 reservoirs functioning, nor on their watersheds. What about residence time in the lakes, fluxes of nutrients and OM from tributaries, activities in the watersheds (soil occupancy...). A table of more dedicated details would be really a key adding.

Reviewer 3 Report

This study presents an investigation on the characteristics of dissolved organic matter (DOM) in five typical eutrophic agricultural reservoirs with different terrestrial pollution sources, which provides meaningful insights into the water-quality management of eutrophic agricultural reservoirs. The subject matter of this study is of great interest. And I think the concepts and methods discussed in this paper are much needed in water resources community for improving water quality management. Thus, I think this paper can be considered for publication after the authors address my major comments raised below.

My main concern is the novelty of this paper. It is more like a technical report rather than a research paper. Even though there is no theoretical contribution, authors are expected to highlight the scientific contribution. Simply investigating the differences in DOM characteristics of the reservoirs with different pollution sources cannot be considered as a significant contribution to the scientific community. All the subsections 2.1-2.4 are too short, with each only having one paragraph. Please increase the length of these subsections. Alternatively, you may combine short subsections into one. Please provide more descriptions about the sampling strategy and physico-chemical analyses so that the experiments can be reproduced successfully. How did you determine the values of fluorescence spectroscopy parameters? What about the quality of data Are there any assumptions and limitations in this study? If any, please clarify.

Round 2

Reviewer 2 Report

Thanks for the modifications You’ll find hereafter my remaining comments (referring to my first comment number):

Comment 1 Global: The English should still be revised. Ok. Ok Regarding DOM its ok. Just take care of a typing error line 39 “andphosphorus”. ok ok. ok Table 1: Regarding “benefit area” I understand your explanation. However, the wording should be improved, or at least you should add an explanation under the table. Table 1: Regarding the details, I would like some more explanations. What do you mean by percentage of pollution line 96-99? Is it a percentage of entering flux or something else? Be so kind to clarify that in your sentences. Regarding the mean residence time, unfortunately your sentence is not clear enough as you provide no data. It means nothing to indicate “a long residence time”. It depends of the involved processes. I understand that you have not data. May be you could at least be able to indicate if it is in the range of months or years. Regarding the other seasons, I don’t agree with your point of view. Practically what occur during the bloom period is totally dependent of the preceding seasons. Therefore, I suggest you add a comment on that in the discussion part of the paper, suggesting a potential interest to look at other seasons in potential future work. Ok Take care of typing error line 108 (reservoir). And please, add in the paper the information you have provided on the hour of sampling (not only indicating during the day). ok ok. Regarding my former comment 14 and your answer, you cannot consider consistency with one other lake to conclude on what occur in your lakes without the data to demonstrate it. It is then only hypothesis based on other publications that should be presented as hypothesis. Moreover, regarding algal succession I understand your answer. I only wanted to say that it could be an essential element to be used for assessing the lakes functioning. The simpler is for you to explain that it could be useful to add complementary insight on algal succession for more detailed assessment. Regarding my former comment 15, I don’t fully agree with your revised text. A low N doesn’t promote growth of algae. It depends of the algal species. I strongly suggest you replace that sentence by a sentence indicating that low N impact the algal succession. Moreover, you indicate that N is the limiting factor. Here again it is wrong. N is an impacting factor, but the disappearance of N in water doesn’t suppress algal growth. Here again it impacts algal species. It is different. Be so kind to rewrite your sentence. Ok ok. ok. ok replace “living source (domestic sewage)” by “domestic sewage” It is sufficient and more understandable. ok. ok. Line 76: What do you mean by "compositional changes"? Line 178: Add the max value of Chla as for the other parameters. Line 232: typing error “>1indicate” Moreover, what do you mean by “realted”? Line 264 twice the same words.

Reviewer 3 Report

The authors have already addressed all my comments and suggestions. I think the revised manuscript can be accepted for publication.

Round 3

Reviewer 2 Report

Thanks for the modifications You’ll find hereafter my remaining comments (referring to my prior comments number):

Point 1 Global: Regarding the English I am not sure to fully understand. Did you use an English native people to amend the paper, or you used an automatic tool for that? Points 4 and 5: Table 1: Let’s say it is OK, even if English should be improved. Point 6: I don't think there is a specific range for reservoirs, even in Korea. It is totally site dependent. Please suppress the end of the sentence "and within the range of Korean eutrophic reservoirs" Point 10: Could you add at the end of your sentence “For clarification, concomitant nutrients and algal succession should be looked for in future complementary assessment.” Point 13: I understand. However, the wording is confusing. Be so kind to replace “compositional changes” by something such as “change of DOM composition” that would be much understandable. Point 14: Many thanks. I just want to take the opportunity of this adding to pinpoint the need to a revision of the English. For instance, the word “profound” is not adapted. Be so kind to revise the English.
